# Oases in the Sahara Desert–Linking biological and cultural diversity

**Laura Tydecks**[1,2], **Juan Antonio Hernández-Agüero**[3¤]*, **Katrin Böhning-Gaese**[4,5], **Vanessa Bremerich**[1], **Jonathan M. Jeschke**[1,2,6], **Brigitta Schütt**[2], **Christiane Zarfl**[7], **Klement Tockner**[3,4]

**1** Leibniz Institute of Freshwater Ecology and Inland Fisheries, Berlin, Germany, **2** Freie Universität Berlin, Berlin, Germany, **3** Senckenberg Gesellschaft für Naturforschung, Frankfurt (Main), Germany, **4** Faculty of Biosciences, Goethe University, Frankfurt (Main), Germany, **5** Senckenberg Biodiversity and Climate Research Centre (BiK-F), Frankfurt (Main), Germany, **6** Berlin-Brandenburg Institute of Advanced Biodiversity Research (BBIB), Berlin, Germany, **7** Department of Geosciences, Eberhard Karls University of Tübingen, Tübingen, Germany

¤ Current address: Department of Environmental Geography, Vrije Universiteit Amsterdam, Amsterdam, The Netherlands
* j.a.hernandezaguero@vu.nl

**Data Availability Statement:** All data is available in Supplementary Material.

**Funding:** JMJ acknowledges financial support from the Deutsche Forschungsgemeinschaft (DFG;

## Abstract

The diversity of life *sensu lato* comprises both biological and cultural diversity, described as "biocultural diversity." Similar to plant and animal species, cultures and languages are threatened by extinction. Since drylands are pivotal systems for nature and people alike, we use oases in the Sahara Desert as model systems for examining spatial patterns and trends of biocultural diversity. We identify both the underlying drivers of biodiversity and the potential proxies that are fundamental for understanding reciprocal linkages between biological and cultural diversity in oases. Using oases in Algeria as an example we test current indices describing and quantifying biocultural diversity and identify their limitations. Finally, we discuss follow-up research questions to better understand the underlying mechanisms that control the coupling and decoupling of biological and cultural diversity in oases.

## Introduction

For the last 20 years, it has been discussed that the diversity of life in a broad sense comprises biological and cultural (incl. linguistic) diversity [1–4], which can together be termed "biocultural diversity" [5] but this is still rarely considered in research approaches. Cultural diversity is tightly related to the resilience of social systems, while biological diversity is pivotal for the resilience of natural systems [6]. Indigenous people and local communities hold in-depth knowledge about species composition and functions as well as ecosystem dynamics [6, 7]; therefore, their knowledge is seen as an insurance for biodiversity, supporting the sustainable management of natural resources [6]. At the same time, natural resources are the basis for the formation of human societies and civilizations, supporting the development of manifold cultures and languages [7].

The biocultural diversity approach, which encompasses the mutual linkages between biological and cultural diversity, examines whether cultural diversity exhibits patterns and

JE 288/9-1, 9-2) < https://www.dfg.de/ >. The funders had no role in study design, data collection and analysis, decision to publish, or preparation of the manuscript.

**Competing interests:** The authors have declared that no competing interests exist.

processes similar to biological diversity and focuses on the co-evolution between human populations and natural plant and animal assemblages [8–13]. The co-evolution of biological and cultural systems has been investigated from various perspectives, for example gene-culture co-evolution, although most of this research describes how nature and culture are being co-produced [14]. The concept of gene-culture co-evolution means that organisms, particularly humans and other key ecosystem engineers [15], shape their environment but also drive evolutionary processes [16]. Indeed, cultural processes may cause genetic adaptations, such as the development of lactose-tolerance through the inception of farming [16]. In addition, the results of cultural processes, such as urbanization, affect the behavior of organisms [17]. Birds, for example, may shift from migratory to sedentary behavior [18], and artificial light at night may reduce the foraging activity of selected bat species despite high food availability [19].

Similar to genes, species, and ecosystems, cultures (incl. languages) undergo cycles of formation, expansion, transformation, and extinction. Cultural extinction is particularly common in regions where ecosystems are degrading and people are marginalized by globalization [20]. Originally, the concept of biocultural diversity focused on traditional and indigenous human societies–particularly in countries of the Global South–and their management and preservation of biodiversity and ecosystem services [13]. Today, it also includes the concept of biocultural creativity in urban contexts, e.g., the formation of novel biodiversity due to management and cultural activities [12, 13]. Hence, a conservation approach integrating biocultural diversity is needed, as already postulated by global organizations such as the United Nations Educational, Scientific and Cultural Organization and United Nations Environment Programme [7], by initiatives like the Millennium Ecosystem Assessment [21] and, more recently, by the Intergovernmental Science-Policy Platform on Biodiversity and Ecosystem Services [3]. Crucially, the methods to describe biocultural diversity are still in their infancy and are often applied only at a country level (e.g., the Index of Biocultural Diversity by [22]).

Drylands are home to unique farming systems, diverse nomadic cultures, and a quarter of the world's languages [23, 24]. Oases play a fundamental role in drylands: they form distinct locations of tight interactions between humans and nature [25, 26] and thus create and maintain biological [27] and cultural diversity. Further, they often contain a unique agrobiodiversity [28], are described as "*in situ* conservation centers for ancient germplasm" [29, 30] and finally provide a wide range of ecosystem services [31, 32]. Consequently, this unique agrobiodiversity facilitates biocultural creativity. At the same time, oases exhibit a long history of human civilization, forming pivotal steppingstones along trade routes and supporting social and economic innovations [27], especially traditional oases [31]. They thereby bridge economies and cultures across geographic and political boundaries (e.g., the Silk Road in western China; [27]).

## Objectives

Due to their characteristics, oases can be considered as ideal model systems for the investigation of drivers of biocultural diversity. Saharan oases show distinct gradients in size, human population density, and connectivity. At the same time, the Sahara Desert, encompassing different countries, exhibits a distinct environmental development (e.g., changing climatic conditions). Thus, the status of biocultural diversity of oases in the Sahara may provide key insights into the underlying mechanisms that control patterns and changes in biocultural diversity. In the present paper, we discuss–from a natural scientific perspective–potential linkages between biological and cultural diversity by identifying relevant drivers and proxies, using traditional oases (*sensu* [31]) in the Sahara Desert as model systems. In the first part, we propose a biocultural concept for oases in the Sahara Desert and discuss the underlying drivers, incl. potential

proxies, for changing biological and cultural diversity. This work is based on an extensive literature review–including scientific literature, project reports, maps, and information derived from sources of international organizations (e.g., UNESCO). In the second part, we examine Algerian oases as a case study in greater detail to test our conceptual framework and investigate the biocultural diversity of oases by referring to and adapting existing methods. Finally, we identify follow-up research questions to advance our understanding of the mechanisms that control the coupling and decoupling of biological and cultural diversity in oases.

## Linking biological and cultural diversity in oases

Mechanisms influencing biodiversity, such as the species-area or species-isolation relationship, are well-studied, in particular for islands (e.g., [33–35]). The study of the cultural facets of human populations in the Sahara Desert also has a long tradition, reaching back to the 18th and 19th century [36]. The variation in both cultural and biological diversity is most likely an interlinked product of historical, ecological, and environmental parameters and drivers [9]. Therefore, a fundamental question is whether cultural diversity exhibits patterns and processes similar to biological diversity, and at which spatial and temporal scales (e.g., [37–39]). To assess this question, drivers of biological and cultural diversity need to be identified. Also, it is important to study the best proxies to be used in the linkages between cultural and biological diversity (Fig 1).

### Drivers of biological and cultural diversity in oases

In general, the dynamics of cultural diversity can be considered from various perspectives. Cultural diversity reflects unique historical events, settlement time in a particular area, and finally cultural diversity is interlinked with environmental conditions such as productivity and temperature [9]. Regarding biological diversity, it is well known that the number of species in

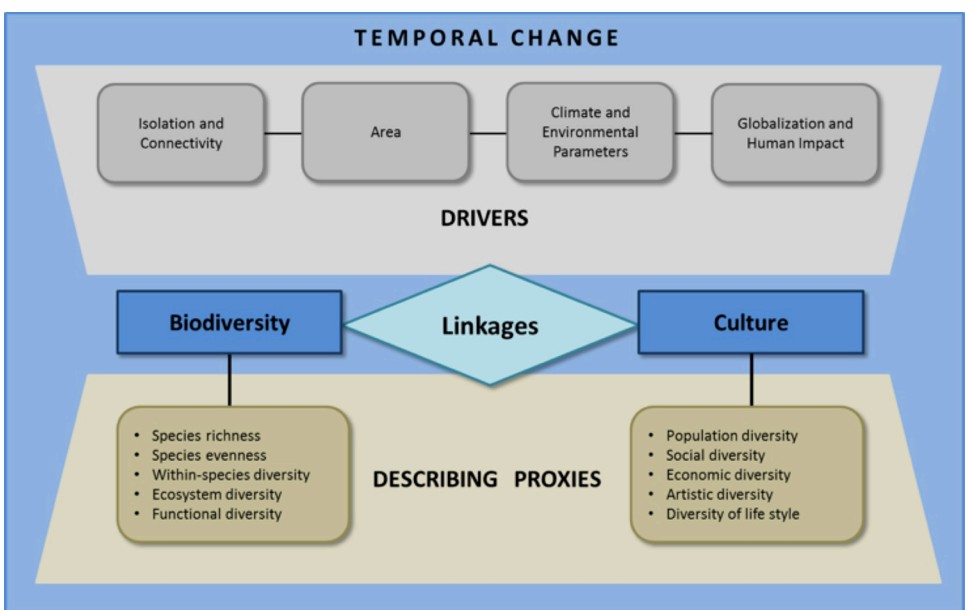

**Fig 1. Schematic presentation of drivers and describing proxies of biocultural diversity.** Proxy groups are listed for both biological and cultural diversity (see section: Proxies for biocultural diversity in oases). Temporal change plays an overarching role as it affects environmental factors (e.g.,. climate and size of oasis), degree of connectivity, and human influences.

a given area varies with latitude, temperature, productivity, spatial extension, and the degree of environmental perturbation [9]. The biocultural diversity of the oases in the Sahara Desert is a consequence of historical and contemporary climatic and environmental conditions, area, different facets of isolation and connectivity, settlement history, as well as of globalization and direct human impacts. The drivers that change the biocultural diversity in Saharan oases are defined here.

## Climate and environmental parameters

During the African humid period, the Sahara Desert was covered by open forests and grasslands. Perennial lakes and rivers allowed for the area to be widely inhabited by humans [40, 41]. Since the end of the African humid period (about 6000 years BP), continuing aridification throughout North Africa created an arid to hyper-arid environment in the Sahara [42]. The present climatic characteristics have existed for approximately 2000 years [43]. In oases, a benign local microclimate developed through an abundance of vegetation, which contrasted with the surrounding arid climate [44], hence creating favorable environmental conditions for both people and nature.

Biocultural diversity found in oases is explained by the presence of water. In deserts, groundwater is recharged by erratic rainfall mostly occurring hundreds of kilometers away. Specifically, most Saharan groundwater resources are of fossil [45], carried to the surface by springs and feeding oases [46, 47] determining the oases' distribution and development in space and time [48]. The dependency on (fossil) groundwater often creates a clustered distribution of oases–similar to an archipelago. Oases appear along geological features such as escarpment ridges and fault lines (e.g., Suegedim-Dirkou-Bilma/Niger), foothills, and wadi riverbeds, where groundwater reservoirs accumulate close to the surface. Fault oases are formed and fed by springs, where groundwater hits an impermeable rock at a fault and ascends to the surface due to hydraulic pressure. Basin oases, on the other hand, typically lie in depressions fed by groundwater flows from the surrounding uplands [45]. Due to their endorheic character, they may form playa (i.e., dry) lake systems under the prevailing arid conditions, locally fed by ephemeral streams [47].

Biological and cultural diversity is hence location-dependent and particularly influenced by climatic and environmental parameters. It has been shown that high latitudes, plains (i.e., areas of low topographic roughness), and dry climates tend to correlate with low biological and linguistic diversity [5, 39]. Nonetheless, drylands are biologically very diverse, with many endemic habitats and species [49]. Species diversity changes with fluctuations in climate, land use, and nutrients [50] and overall species richness might be lower in drylands than in tropical forests, but within-species diversity is very pronounced due to the diversity and isolation of habitats and associated populations [49].

## Area

The area of oases varies over several orders-of-magnitude and depends on climate, geological and topographical settings as well as human impacts (details below). For example, the archipelago of the "Oases du Kawar" (Niger) encompasses an area of 3685 km$^2$ while the "Oasis de Ouled Saïd" (Algeria) and the "La Vallée d'Iherir'" (Algeria) encompasses an area of 254 km$^2$ and 65 km$^2$, respectively (https://rsis.ramsar.org/). In general, area is the strongest predictor of species richness, particularly on islands [35]. Large island areas harbor more individuals, because of the higher provision of resources and energy [33, 34].They contain a greater variety of habitat types, thereby increasing local and regional species diversity. However, the number of habitat types is not solely controlled by area but also by the degree of landscape

heterogeneity. Furthermore, large island areas have a higher potential for *in situ* speciation [34], experience lower extinction rates, and tend to accumulate endemic species [51]. In relation to cultural diversity, the richness of languages has been observed to increase with area [22], as a larger area can support a larger human population.

## Isolation and connectivity

Isolation is a fundamental attribute of islands [10, 52] that can also be applied to island-like systems such as oases [53]: oases can be seen as "vegetated islands" within drylands. Isolated islands are, in general, less affected by cultural and ethnic change, hostile invasion, mass immigration, or political interference. At the same time, they are exposed to cultural input from a wide range of sources [52]. Oases may experience a similar development of cultural input through their position along trade routes. Furthermore, the degree of isolation is an important factor for biodiversity and its change. Isolation as well as the variety of habitat types might jointly cause a higher within-species diversity in oases [49], and this is explained by (i) spatial, (ii) geological, (iii) historical, and (iv) modern connectivity.

i. *Spatial and landscape connectivity* encompasses the distance to the borders of the Sahara, which influences the biological and cultural diversity of oases. Oases close to the northern margin of the Sahara may be impacted by the Mediterranean landscape, whereas the southern oases are primarily influenced by the Sahel zone [54]. Furthermore, oases distributed throughout the Sahara Desert might serve as steppingstones, connecting the various parts of Africa. These steppingstones are, for example, essential for migratory birds [23, 49] but also for nomadic and dispersing people [49] (e.g., Saharan Gold Route and Asian Silk Road; [55]).

ii. *Geological / hydrological connectivity* leads to the formation of neighboring oases (archipelago) through shared aquifers. Geographic and tectonic development may generate specific patterns of species diversity through time [50]. Hence, a shared aquifer in the Sahara might support similar (aquatic) species compositions of oases. Indeed, spatial proximity facilitates both cultural and biological exchange among oases.

iii. *Historical connectivity* can be described through trade and travel routes. Today, two thirds of the Saharan human population are sedentary and live in oases [56], which provide important farming areas and stops along trade routes (Fig 2). Trans-Saharan trade has existed since prehistoric times, peaking from the 8th to the late 16th century [49]. Therefore, oases are cultural hotspots, desirable to political, economic, and military leaders [57]. Furthermore, migratory corridors used by caravans and herding groups enabled long-distance dispersal of plants and animals (as well as cultures) from oasis to oasis, which otherwise are embedded in an unsuitable landscape matrix. Through the provision of fodder, water, and shade, biological and cultural diversity has spread along these routes. The ongoing use of these trade routes by livestock caravans ensures the preservation of both cultural and biological diversity. For example, numerous cultures and a great diversity of cultural heritage sites emerged along these trade routes [49].

iv. *Modern connectivity* is defined by the connectedness of oases through roads, airports, nearest towns, and inhabited neighboring oases. In general, exchange among human populations is fundamental for cultural development. The crossing of borders stimulates cultural innovation, while cultural homogenization is supported by boundaries [10]. At the same time, the introduction of modern technology seems to be a major threat for both biological and cultural diversity by providing access to previously almost inaccessible oases with unique biological and cultural features and by promoting tourist activities [59].

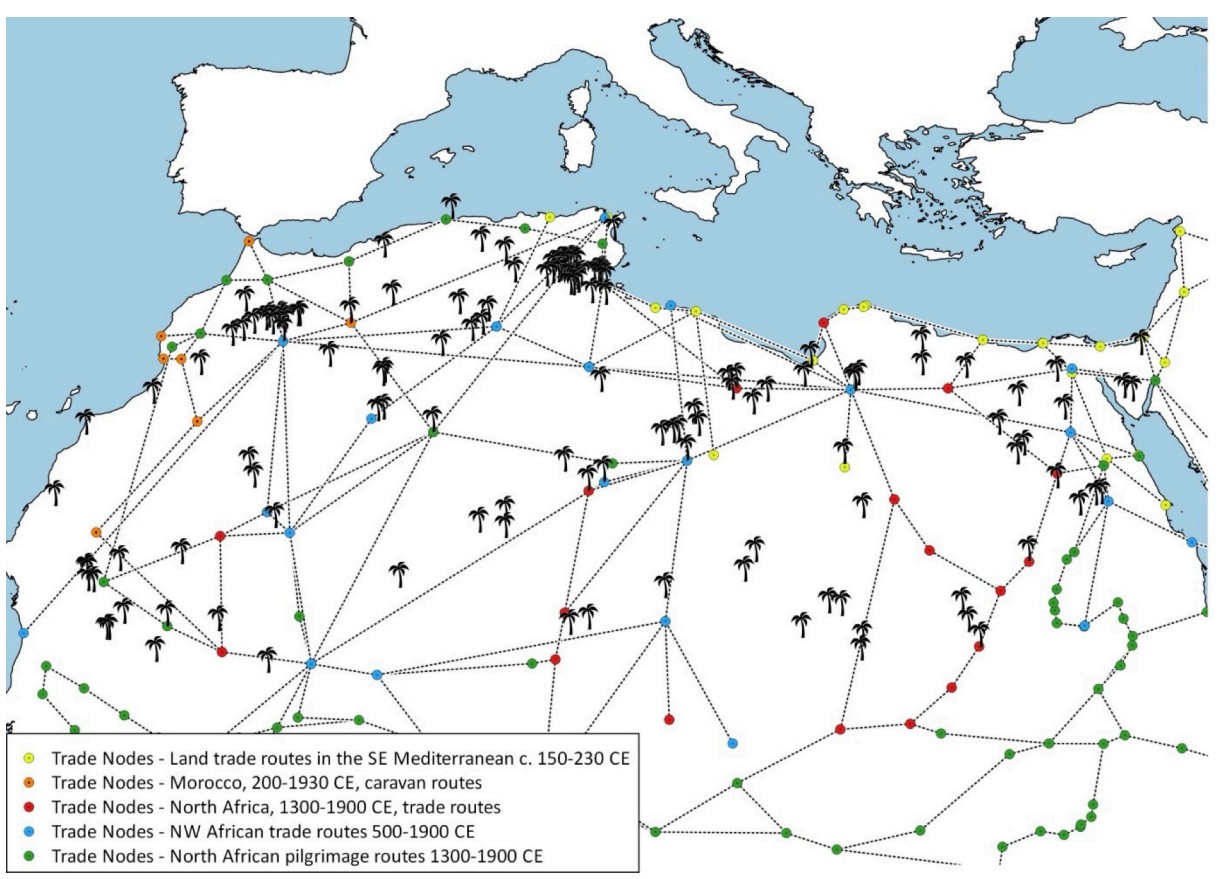

**Fig 2. Oases (palm tree) and different trade routes in the Sahara Desert.** Individual oasis settlements are grouped based on their location in the same valley, depression, or region (compare S1 File: Table S1). The trade routes represent the pathways between stop-over sites (trade nodes, such as marketplaces). Trade routes are used for the transport of cargo and provide a link between producers and buyers (based on Old World Trade Routes Project: http://www.ciolek.com/owtrad.html [58]). The map shows that oases are often located along trade routes and form important trade nodes along these routes.

## Settlement history

After the arid to hyper-arid Ogolien resp. Kamensien during the North European Last Glacial Maximum, the Sahara Desert experienced an expansion, extending southward into the modern Sahel zone [60]. Historically, four occupation phases can be distinguished for the Sahara [40]. First, human settlement started in the reoccupation phase (8500 to 7000 B.C.E.) due to monsoon rains forming a habitable savannah-like environment in the Eastern Sahara [61]. Humans from the South, who were adapted to savannah systems, shifted northward to extend their living area [40]. Second, human settlements were established throughout the Eastern Sahara in the mid-Holocene formation (7000 to 5300 B.C.E.; [40]). Domestic livestock such as sheep, goats, and cattle were introduced [62]. In the final stage of this period, multi-resource pastoralism was the subsistence strategy within the Egyptian Sahara. Third, the Mid-Holocene regionalization (5300 to 3500 B.C.E.) exhibited a sporadic occupation of the western part of the Sahara, while in the eastern part cattle pastoralism and the cultivation of wheat and barley developed [40]. And fourth, long-distance migration through the hyper-arid Sahara relied on donkeys in the late Holocene marginalization (3500 to 1500 B.C.E.; [61]), as camels had not yet been introduced [40]. Due to the high water needs of donkeys, geographical knowledge and mobility were crucial for survival. In this period, humans changed their way of living from

foraging to a multi-resource economy and to specialized pastoralism [40]. This unique settlement history, strongly linked to climatic conditions, is considered a key driver influencing biological and cultural facets in oases.

## Globalization and human impact

Human colonization affects linguistic evolution and forces cultural homogenization [39]. As Africa in general has been strongly influenced by colonialism, African oases have experienced a similar impact, as seen, for example, in the M'zab Valley in Algeria [63]. At the same time, globalization processes such as trade and investment flows, transportation and telecommunication, and cultural exchange, have reduced insularity and increased the homogeneity of languages. On average, two languages go extinct every month worldwide [10]. Oases are strongly impacted by human activities [26] and globalization: traditional agricultural systems are increasingly replaced by modern systems, modern agricultural projects are initiated by governments, and tourism is increasing. Consequently, tomatoes and other vegetables are produced in addition to the traditional produce (dates and olives) and delivered to hotels and restaurants (e.g., in the Siwa Oasis, Egypt; [64]). With growing human population in oases, local adaption strategies to water scarcity have been developed, while ongoing population growth may have resulted in the exploitation of groundwater resources to meet the people's often-conflicting economic goals (e.g., agriculture and tourism). Climate change and exhaustion or abuse of water resources may have led to the complete disappearance of oases [27]. However, information for the Sahara Desert on disappearing oases is not available. Other oases exhibited a change in size and structure through human population expansion, agricultural development, and the rise of tourism [27].

## Proxies for biocultural diversity in oases

While patterns and dynamics of biological diversity have been reported in numerous studies, cultural diversity as such is not routinely measured and is difficult to explore and generalize. Based on the previous discussion of drivers of biocultural diversity and on our review of relevant literature on oases and the biocultural diversity approach, we briefly assess important proxies for biological and cultural diversity in oases. To provide an overview of what information is needed from our perspective to attain an understanding of linkages between cultural and biological proxies and the underlying drivers, we gathered exemplarily detailed information for the Siwa Oasis in Egypt (S1 File: Table S2), for which we have a sound data basis.

## Proxies to describe biological diversity in oases

In the context of the existing biocultural concept, biological diversity is mainly represented by species richness, in particular of mammals, plants, and birds (e.g., [22]). However, deserts are known to also have a high diversity of amphibians and reptiles [24, 65]. Furthermore, within-species diversity in oases is known to be very high, particularly for agrobiodiversity [28, 66, 67]. To understand how climatic and environmental parameters, area, connectivity, and temporal development in and of oases influences biodiversity, we therefore propose to measure biological diversity in greater detail than suggested by [22]. Specifically, we propose the following five proxy groups to ideally describe biological diversity: (a) species richness, including agrobiodiversity, endemic species, non-native species, endangered and extinct species; (b) species evenness; (c) within-species diversity, including genetic and phenotypic diversity, comprising behavioral diversity; (d) ecosystem diversity, including habitat richness; and (e) functional diversity. However, some of these proxy groups are difficult to underpin with

empirical data. Therefore, we give detailed information on proxies and their potential implementation related to the proxy groups.

Species richness can be derived from the number of different species occurring in an oasis based on occurrence records from GBIF (http://www.gbif.org/), BirdLife data (http://www.birdlife.org/), and other online and literature sources. The IUCN Red List of Threatened Species (http://www.iucnredlist.org) gives further information on endangered and extinct species. For oases, agrobiodiversity is relevant, in particular, indicating a strong connection between biological and cultural diversity. Hence, the number of different cattle and crop species and varieties should be identified for oases. Date palm oases form key agroecosystems in drylands and are important for protecting genetic resources as well as providing habitats and refugia for faunal diversity [67–69]. Hence, date palms are a very important fruit tree, with high economic importance, in oases. In addition, they show high genetic diversity (e.g., [70, 71]), which makes them an ideal proxy for describing within-species diversity in oases (compare Siwa Oasis, S1 File: Table S2). Data on species evenness, ecosystem diversity, and functional diversity are not yet available for most oases.

## Proxies to describe cultural diversity in oases

Culture not only evokes images of art and fashion, but comprises beliefs, values, behavior, and traditions associated with a particular human population [16]. In the context of the biocultural concept, the number of languages, ethnic groups, and religions is mainly used to describe cultural diversity [5, 22]. Language diversity is the most frequently used proxy for cultural diversity because languages encode collective knowledge in a way that is often non-translatable [6]. For a more comprehensive view of cultural facets in oases, however, we propose an extension of those indicators and the consideration of five proxy-groups (in part proposed by [72]): (a) human population diversity, including ethnic groups, (endangered) languages and religions; (b) social diversity, including the type of social organization; (c) economic diversity, including economic sectors and their weighting; (d) diversity of life styles, including specific diets, important celebrations, and land management practices; and (e) artistic diversity, including specific architecture, traditional crafts, and music (exemplarily illustrated for Siwa Oasis, S1 File: Table S2).

Human population diversity can be described by the number of ethnic groups, number of languages and endangered languages, and the number of religions. This data can be derived from the Ethnologue database on languages of the world (https://www.ethnologue.com/), the UNESCO Atlas of the World Atlas of Languages in Danger (http://www.unesco.org/languages-atlas/), the Joshua project (https://joshuaproject.net/) and the World Christian Database (http://www.worldchristiandatabase.org/wcd/home.asp) which may be further complemented from additional sources such as relevant literature, in particular. The remaining proxy groups describing cultural diversity are difficult to assess quantitatively. Instead, they can complement the comprehensive view on oases in a qualitative way. Such information may be derived from individual literature sources.

## Linkages between proxies for biological and cultural diversity in oases

The independently developed proxies describing biological and cultural diversity are linked through tight coupling of humans and nature in oasis systems. Here, drivers of biological and cultural diversity are shaping proxies of both biological and cultural diversity and may form an interdependency of both. For example, environmental and climate parameters fostered the development of oases in the past, offering a habitable environment for humans in an otherwise hyper-arid surrounding. On the other hand, the developed proxies may be linked

directly, e.g., different ethnic groups are maintaining different biological diversity. This means that the "diversity of ethnic groups" is both a *proxy* for cultural diversity as well as a *driver* for biological diversity. In addition, we may observe direct feedback between a proxy and a driver, e.g., the driver "land use" may be directly impacted by the proxy "species richness." This causes a high non-linearity within the interdependencies of drivers and biological and cultural proxies and will have to be considered in further, more quantitative analyses. Also, to identify and understand direct linkages between proxies and the degree of importance of the identified drivers, further studies need to be conducted.

## Case study–Algerian oases

For a detailed case study and feasibility analysis, we collected information on oases in Algeria (in total: 77 oases, 18 oasis groups), partially testing our conceptual framework outlined above. We use area, human population, and connectivity (expressed as geographic distance) as drivers and consider languages and ethnic groups as cultural proxies and species richness as a biological proxy. Here, we provide an analysis of contemporary Algerian oases, ignoring temporal trends.

We describe the biocultural diversity in each oasis (or oasis group) and compare this diversity among oases. Two complementary methods are applied: (1) the pairwise comparison of oases, or of oases groups, using the Jaccard Index (JI) [73], and (2) the expression of the overall biocultural diversity in each oasis (or group), using the Index of Biocultural Diversity (IBCD), developed by [22], and testing the influence of area and human population, respectively. With the JI (method 1), we investigate the correlation between spatial proximity of oases and the similarity in their cultural and biological diversity. The larger the JI, the more similar are the oases. We hypothesize that biological and cultural dissimilarity increases with the geographic distance between oases. With the IBCD (method 2), we compare the overall biocultural diversity in the oases. A larger IBCD indicates greater biocultural diversity. Adjusted for area and human population, we test the influence of these two drivers on biocultural diversity. Following [22], who calculated the IBCD for individual countries, we hypothesize that there is a positive relationship between area and cultural and biological diversity, and between human population and cultural diversity in oases. In addition, we hypothesize that more populous oases tend to have a greater biological diversity than less populous ones because of the importance of traditional agriculture on biodiversity. Finally, we hypothesize that there is a positive correlation between the number of languages and number of species in Algerian oases, as already demonstrated for other systems (e.g., [37, 39, 74, 75]).

### Methods to conduct the case study

To create a list of Algerian oases, bibliographic research was conducted in Google Scholar and Google Browser in March 2017, including scientific literature, project reports, maps, and information derived from sources of international organizations (S1 File; Table S1).

Information on location (for 70 of 77 oases), area (for 47 oases), population (for 53 oases), cultural aspects (ethnic groups, languages, endangered languages, religions), and biological aspects (species richness) was collected for oases in Algeria. Cultural aspects were gathered using information from Ethnologue (https://www.ethnologue.com/), UNESCO (http://www.unesco.org/languages-atlas/), the Joshua project (https://joshuaproject.net/) and the World Christian Database (http://www.worldchristiandatabase.org/wcd/home.asp). Species richness for each oasis was calculated using the intersection tool in QGIS [76]. As only point locations for each oasis were available instead of polygons representing the shape of the oases, we used a buffer of 20 km around each oasis point. Occurrence data and distribution maps from BirdLife

[77], IUCN [78] and GBIF data [79] on species level were used. We calculated the Jaccard Index (JI) for each oasis pair and the IBCD for each oasis. As oases are often clustered and form an archipelago, we also conducted the analyses on an oasis group level. With this approach, we can gain a better understanding of the role of geographic distance among oases and may receive insights into external drivers such as area or environmental parameters.

Based on presence/absence data, we calculated the JI on an individual oasis level as well as on an oasis group level. We considered pairwise similarities between oases using a binary matrix. This comparison of oases and oasis groups excludes double zeros, as the absence of species may be caused by various factors and therefore does not necessarily reflect differences in the environment or cultural proxies [80]. However, the JI, to a certain degree, describes beta diversity, defined by [81] as the "extent of change in community composition among sites". All terms are weighted equally, meaning that rare biological proxies (e.g., endangered species) and rare cultural proxies (e.g., endangered languages) have equal weight. As JI expresses the dissimilarities between a pair, we calculated the similarities by subtracting JI from 1 (in the following we use the term JI to express this value).

To test the influence of proximity between two oases (or oasis groups), we calculated the correlation between JI and distance and visualized it as scatterplots. JI was calculated using R [82]. The distance between oases was calculated using ArcGIS [83].

We adjusted the approach of [22] for an index of biocultural diversity (IBCD) to our case study. We calculated the index for both individual oases and oasis groups and adjusted it for area and population, following the equations of [22]. Mammals, birds, amphibians, and reptiles are known to be important biological proxies in deserts [24, 65], and invertebrates are critical components of biodiversity as well. We therefore assessed faunal diversity in greater detail than in the original IBCD approach and included birds, mammals, amphibians, reptiles, arthropods, and mollusks. Furthermore, on a country level the cultural proxy "religion" is known to be the most influential factor because of one dominant religion in most countries [84]. We discovered this pattern in Algerian oases as well, where Islam and Christianity are the dominant religions. Therefore, we excluded "religion" in the calculation of JI and IBCD.

## Results of the case study

The area of Algerian oases ranges from 2 km$^2$ (Ouargla) to 211,980 km$^2$ (Beriane, located in the M'zab Valley; S1 File: Table S3). Human population ranges from 1267 inhabitants (Tamtert, Saoura Valley) to 215,000 inhabitants (Biskra, an oasis city located in the archipelago of Ziban; S1 File: Table S3).

A total of 552 plant species, 14 amphibians, 150 arthropods, 328 birds, 98 mammals, and 72 reptiles have been found in Algerian oases. Furthermore, 12 ethnic groups and languages, incl. five endangered languages (spoken in 12 oases), can be found in Algerian oases. Eleven out of 78 oases are situated within a Ramsar site, and the M'zab Valley with its seven oases (cultural site) and Tas'ili' n'Ajjer (cultural and natural site) are listed as UNESCO World Heritage Sites. Wilaya d'El Oued is a "Globally Important Agricultural Heritage System" (GIAHS), representing the Ghout System, a traditional hydro-agricultural system (S1 File: Table S3).

On an individual oasis level, and as hypothesized above, there is a significant correlation between the number of species and languages (r = 0.782, p < 0.001). In contrast, there is no significant correlation on an oasis group level (r = 0.344, p = 0.163). On an individual and group level, there are no correlations between biological and cultural proxies, respectively, and area. In contrast, all proxies correlate with human population size (both levels; Table 1). No significant correlations between biological and cultural proxies, respectively, and the distance to the nearest oasis and oasis group exist (both levels; Table 1A and 1B).

**Table 1. Correlations between (i) area, human population, and distance and (ii) biological and cultural diversity proxies on the level of (A) individual oases and (B) oasis groups in Algeria.** Significant correlations are in bold.

| | Area | | | Human population | | | Distance | | |
|---|---|---|---|---|---|---|---|---|---|
| **(A)** | **corr** | **p-value** | **df** | **corr** | **p-value** | **df** | **corr** | **p-value** | **df** |
| **Biological Diversity** | -0.191 | 0.199 | 45 | **0.377** | **0.005** | 51 | -0.166 | 0.180 | 65 |
| **Fauna** | -0.196 | 0.187 | 45 | **0.326** | **0.017** | 51 | -0.213 | 0.084 | 65 |
| **Plants** | -0.158 | 0.290 | 45 | **0.412** | **0.002** | 51 | -0.068 | 0.583 | 65 |
| **Cultural Diversity including Religion** | 0.117 | 0.434 | 45 | **0.600** | **0.002** | 51 | **-0.275** | **0.024** | 65 |
| **Cultural Diversity excluding Religion** | 0.104 | 0.485 | 45 | **0.588** | **0.004** | 51 | -0.213 | 0.083 | 65 |
| **Languages** | 0.102 | 0.494 | 45 | **0.450** | **0.001** | 51 | -0.236 | 0.055 | 65 |
| **Religions** | 0.113 | 0.451 | 45 | **0.320** | **0.019** | 51 | **-0.296** | **0.015** | 65 |
| **Ethnic Groups** | 0.089 | 0.552 | 45 | **0.646** | **0.000** | 51 | -0.143 | 0.248 | 65 |
| **Biocultural Diversity including Religion** | -0.187 | 0.208 | 45 | **0.386** | **0.004** | 51 | -0.170 | 0.169 | 65 |
| **Biocultural Diversity excluding Religion** | -0.188 | 0.206 | 45 | **0.385** | **0.004** | 51 | -0.168 | 0.174 | 65 |
| **(B)** | | | | | | | | | |
| **Biological Diversity** | 0.149 | 0.554 | 16 | **0.793** | **0.001** | 11 | -0.067 | 0.791 | 16 |
| **Fauna** | 0.205 | 0.415 | 16 | **0.787** | **0.001** | 11 | -0.013 | 0.960 | 16 |
| **Plants** | 0.020 | 0.937 | 16 | **0.723** | **0.005** | 11 | -0.168 | 0.505 | 16 |
| **Cultural Diversity including Religion** | 0.421 | 0.082 | 16 | **0.677** | **0.011** | 11 | -0.067 | 0.791 | 16 |
| **Cultural Diversity excluding Religion** | 401 | 0.099 | 16 | **0.632** | **0.020** | 11 | -0.086 | 0.734 | 16 |
| **Languages** | 0.411 | 0.090 | 16 | **0.586** | **0.035** | 11 | -0.109 | 0.667 | 16 |
| **Religions** | 0.424 | 0.079 | 16 | **0.652** | **0.016** | 11 | -0.014 | 0.956 | 16 |
| **Ethnic Groups** | 0.341 | 0.166 | 16 | **0.572** | **0.041** | 11 | -0.050 | 0.844 | 16 |
| **Biocultural Diversity including Religion** | 0.156 | 0.535 | 16 | **0.796** | **0.001** | 11 | -0.068 | 0.789 | 16 |
| **Biocultural Diversity excluding Religion** | 0.154 | 0.541 | 16 | **0.796** | **0.001** | 11 | -0.068 | 0.788 | 16 |

One of the most similar oasis pairs, as described by JI, is Biskra and Biskra-El Ghrous within the oasis group Ziban (JI = 0.886), located 18.5 km apart. Among others, Bskra—Ouled Djellal and Zaouiet Kounta exhibit the highest dissimilarity (JI = 0.002), located 980 km apart (S1 File: Table S4). On an individual level, our results support the hypotheses that biological and cultural similarity increases with decreasing distance between oases (S1 File: Table S5, Fig 3). Among groups of oases, this is only true for languages (S1 File: Table S5, S1 Fig). The most similar oasis group pair is d'Ideles and Wilaya de Tamanrasset (JI = 0.453), located 79.6 km apart, whereas Djanet and Oasis de Moghrar et de Tiout exhibit the highest dissimilarity, located 1262 km apart (JI = 0.004; compare Table S6 in S1 File).

The bioculturally most diverse oasis in Algeria is Biskra (0.616; IBCD-RICH). Adjusted for area and to human population, the bioculturally most diverse oases are Ghardaia (0.610; IBCD-AREA) and Alamellal (1.034; IBCD-POP), respectively. The culturally most diverse oasis is Adrar (CD = 0.5), while several oases (e.g., El Golea or Tamtert) are tied for the culturally least diverse (CD = 0.0; Table S7 in S1 File). The biologically most diverse oasis is Biskrai—Ouled Djellal (BD = 0.833), while Zaouiet Kounta is the biologically least diverse (BD = 0.053) (Table S7 in S1 File). On an individual level, there is no correlation between BD-RICH and CD-RICH (Table S8A in S1 File).

The bioculturally most diverse oasis group is Ziban (0.648 for IBCD-RICH; 0.753 for IBCD-AREA) and the group Tiout, adjusted for population (0.996 for IBCD-POP; RAMSAR; Table S9 in S1 File). The culturally most diverse oasis group is Tuat (CD = 0.5), while El Golea (CD = 0.0) is the culturally least diverse group. The biologically most diverse group is Ziban (BD = 0.853), and the Oasis de Tamantit et Sid Ahmed Timmi (BD = 0.087; RAMSAR site) is the biologically least diverse oasis group (Table S9 in S1 File). On a group level, there is a

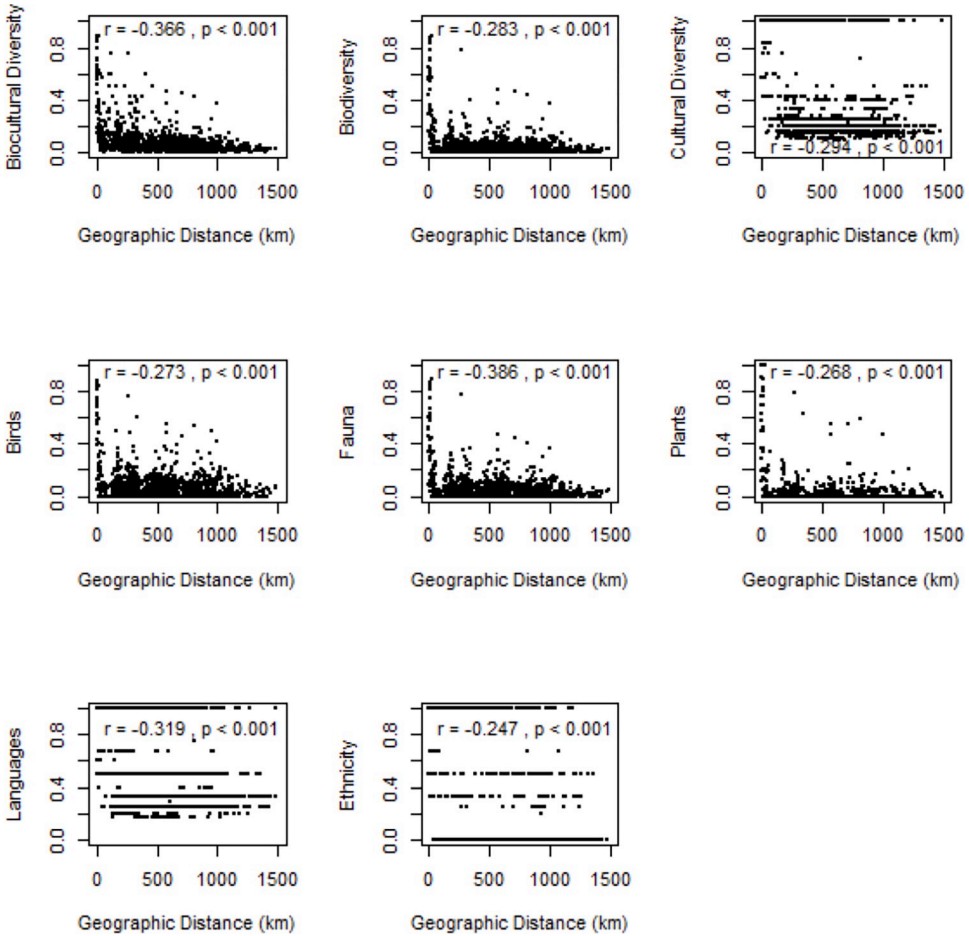

**Fig 3. Relationships between similarity in biocultural diversity.** Relationships between similarity in biological and cultural diversity between individual oases in Algeria (different proxies given on y-axes) and their geographic distances (x-axes). Pearson correlation coefficients (r) and significance values (p) are shown.

significant and positive correlation between BD-RICH and CD-RICH (r = 0.557, p-value < 0.05; Table S8B in S1 File).

## Discussion of the case study

The results of the case study show that Algerian oases are bioculturally diverse systems. Globally, Algeria is on rank 95 out of 221 countries for IBCD-RICH (0.476), on rank 172 adjusted for area (IBCD-AREA = 0.326), and on rank 189 adjusted for population (IBCD-POP = 0.377) [22]. While Algeria is thus considered only a moderately diverse country, the oases are pivotal for its cultural and biological diversity. For example, 12 (based on the present results) out of the 22 languages in Algeria (see [22]) are spoken in oases. Furthermore, [85] report the decline of ten threatened large-size vertebrates in the Sahara-Sahel region due to regional conflicts. Eight of these ten species occur in the Algerian oases. Indeed, these species are all found in the oasis group Ziban, which is considered the bioculturally most diverse group in the entire country. At the same time, 18 out of 77 oases are located in protected areas, emphasizing that the Algerian oases are unique and critical for the conservation of the threatened Saharan megafauna, but also for the cultural diversity, including endangered languages. At the same time, Algeria is listed as one of the most underfunded countries for biodiversity conservation

globally [86], which raises concern about the sustainable conservation of its unique biocultural heritage. Over many generations, indigenous economies and their traditional and local management practices enabled the protection of biological diversity [6, 39]. At present, it remains an increasing challenge to protect the unique biological and cultural diversity in oases, threatened by modernization processes and environmental degradation. Therefore, it is crucial to protect not only biological diversity but also cultural diversity. To reach a balanced protection of biological and cultural diversity, it is important that the protection of cultural diversity is not only seen as a means to protect biodiversity [6]. Furthermore, to support sustainable conservation efforts, the underlying mechanisms of long-term alterations of the functional linkages between biological and cultural diversity must be assessed and understood.

In contrast to our hypothesis, there was no correlation between the area of an oasis and its species richness. Human population size seems to mask such a correlation, emphasizing a close linkage between humans and nature in Algerian oases [28, 69, 87], partly due to unique agricultural systems such as the Ghout System, a GIAHS. Our results therefore provide a first hint at a co-evolution between cultural and biological diversity (based on proxies) in Algerian oases, where people are crucial for maintaining biological diversity. However, the unique agrobiodiversity in Saharan oases is increasingly threatened by the mass introduction of industrial crops [28]. At the same time, it is further jeopardized by increasing tourism [64] and mining activities (e.g., oil, uranium).

Geographic distance between individual oases, rather than between groups of oases, was significantly correlated to biological and cultural diversity. This underpins our hypothesis that clustered oases are biologically and culturally very similar, as demonstrated by the frequent similarity of languages spoken in nearby oases. The language-species relationship in Algerian oases is in line with previous studies (e.g., [37]) and shows that cultural diversity exhibits patterns similar to biological diversity.

Generally, biocultural studies mostly apply proxies on a country level (e.g., [22]), while local and regional studies are rare (e.g., [10]). Different spatial levels (i.e., individual oasis or oasis groups), however, provide different results on biocultural diversity in Algerian oases, emphasizing the need of a careful consideration of spatial scales.

Using IBCD and JI has certain restrictions and could be further developed for future studies. IBCD assumes that cultural proxies follow similar patterns and trends as biological proxies [72]. A key challenge of the present study was the lack of qualified data and information for the oases in the Sahara Desert. Consequently, we could only extract parts of our 'proxy wish list' presented above. In addition, JI considers only pairwise beta diversity and does not provide information on the underlying phenomena such as nestedness (i.e., oases with smaller numbers of species are subsets of oases with higher numbers of species) and turnover (replacement of species by other species). Hence, multiple-site comparisons, including nestedness and turnover, are considered important for future studies [80, 88]. In addition, JI is not adjusted for area, which might cause biased results [88]. Lastly, we lack comprehensive data and information to understand and quantify the temporal dynamics of biocultural diversity in the Sahara desert.

We considered a part of our conceptual framework in the case study and referred to existing methods describing facets of biocultural diversity. In doing so, we provided initial important results of describing biocultural diversity in Algerian oases and hints at understanding the potential co-evolution of biological and cultural diversity.

## Conclusions and outlook

In this paper, we outlined a concept of biocultural diversity based on and applied to oases, including an in-depth examination of drivers and proxies describing and influencing

biological and cultural diversity. However, we are aware that the implementation of such a framework remains a major challenge, as demonstrated in the present case study on Algerian oases. At the same time, understanding the spatial and temporal dynamics of biological and cultural diversity is considered a prerequisite for managing oases in the Sahara Desert in a sustainable way and for maintaining them as biocultural heritages of global importance.

## Protecting biocultural diversity

With increasing efforts to conserve biodiversity, different approaches and strategies were developed over time (e.g., Ecosystem Services, Nature's Contribution to People; [89–92]). An important direct measure for biodiversity conservation is the designation of protected areas. The role of human communities close to conservation areas is being intensively discussed in this context, from proponents of "people-centered conservation" to human-free "fortress-centered conservation" [2, 93]. However, conservation is always linked to social-political contexts [1]. Biodiversity loss, for example, is not only influenced by societal processes but can by itself impact societies [94]. Also, social concerns (e.g., health, economics, education) are typically of higher importance for policy makers than are environmental concerns [95]. Therefore, besides biodiversity conservation as the main goal in protecting areas, the expectation nowadays is to socially and economically contribute to human society as well; for example, contributions toward the livelihood of local communities and national economies (e.g., through tourism) [90].

The concept of biocultural diversity is a promising approach for a holistic and interlinked view of nature and humans. It might function as a complementary approach in conservation by offering a comprehensive view on temporal and spatial scales of biological and cultural diversity, underlying drivers, and relevant proxies. Understanding the co-evolution of biological and cultural diversity offers a holistic picture of interlinkages between biological and cultural proxies and drivers. However, we have shown that there are critical data gaps when applying such a biocultural approach in oases and beyond. Methodological advancements are also needed to better describe biocultural diversity and to find common characteristics.

## Future research needs

Studies on oases have typically focused on hydrology, archaeology, agriculture, or species distributions. However, limited knowledge exists on the coupling and decoupling of biological and cultural diversity, although oases–similar to mountain ranges, islands, and urban areas–constitute ideal model systems for studying the underlying mechanisms influencing these mutual linkages. Unstable political conditions (e.g., rebel activities) constrain research in many oases of the Sahara Desert [85, 96]. In the following, we raise important research questions that future research needs to address in order to achieve a better understanding of the mutual linkage between biological and cultural diversity, using oases as model systems.

Oases form key nodes of biocultural networks (Fig 2), and we have demonstrated that the degree of connectivity is fundamental for biocultural diversity: *How does the degree of connectivity between and among oases influence biological and cultural diversity*? And: *How do changes in the degree of connectivity shape cultural and biological diversity*?

Biodiversity is predicted to increase with area and ecosystem heterogeneity. For biocultural diversity in Algerian oases, however, there is evidence that human population size rather than area is the driver. *How do area, heterogeneity, and human density influence cultural and biological diversity in oases*?

Water is the limiting resource in oases, and water abstraction has been increasing since the middle of the 20<sup>th</sup> century due to rapid demographic and economic development. Weak

governance and a lack of maintenance work of traditional as well as of newly implemented water management schemes have triggered environmental degradation, including soil salinization and loss of vegetation [23]. Hence, a comprehensive understanding of the water budget of individual oases, and of entire archipelagos, is urgently needed, as well as an understanding of the traditional adaption strategies to cope with water scarcity in various oases: *How do water limitations control both biological and cultural diversity and its reciprocal linkages*?

A distinct correlation between land-degradation, loss of biodiversity, and overprinting of traditional/local culture by globalization is a common phenomenon in drylands [49]. Therefore, the "modernization" of oasis systems–corresponding to globalization processes–is considered a threat to biodiversity [68]; at the same time, it may also harm the cultural diversity in the oases. In the Maghrebian oases, for example, the increase in human population, in combination with governmental policies to settle nomadic populations, is considered a key driver of drylands [23]. Therefore, modernization processes have an impact on environmental and cultural parameters: *How do modern developments impact the biocultural diversity in oases*? And: *Are there "keystone" oases that require specific conservation and management strategies*?

As outlined above, during the Holocene the oases of the Sahara underwent strong alterations, corresponding to climate change and settlement activities: *How did these "historic" factors influence biological and cultural diversity*?

In general, an interdisciplinary approach is crucial to generating a more systemic and holistic understanding of oasis systems: *What are the factors determining cultural and biological diversity within individual oases*, *and what is the role of the oases for larger regions*, *the entire Sahara*, *and even beyond*?

Finally, the concepts and models derived from oases may be transferred and applied to other ecosystem types, including mountain ranges, urban areas, and islands, where biological and cultural diversity are coupled and decoupled.

## Supporting information

**S1 Fig. Relationship between biological and cultural diversity and geographic distances in oases groups in Algeria.** Pearson correlation (r) and significance (p) are shown.
(DOCX)

**S1 File.**
(XLSX)

## Acknowledgments

We thank Joanna Mitchell and Merve Ören for collecting data on oasis settlements in the Sahara Desert. We are very grateful to two reviewers and the Editor for providing very helpful comments that considerably improved the text.

## Author Contributions

**Conceptualization:** Laura Tydecks, Katrin Böhning-Gaese, Jonathan M. Jeschke, Brigitta Schütt, Christiane Zarfl, Klement Tockner.

**Data curation:** Laura Tydecks.

**Formal analysis:** Laura Tydecks, Vanessa Bremerich, Christiane Zarfl.

**Investigation:** Christiane Zarfl.

**Methodology:** Laura Tydecks, Vanessa Bremerich, Christiane Zarfl.

**Software:** Vanessa Bremerich.

**Validation:** Vanessa Bremerich.

**Visualization:** Laura Tydecks, Vanessa Bremerich.

**Writing – original draft:** Laura Tydecks.

**Writing – review & editing:** Juan Antonio Hernández-Agüero.

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
