## [Decision Letter · Decision Letter 0]

29 Jun 2023

PONE-D-23-03425Oases in the Sahara Desert – Linking Biological and Cultural DiversityPLOS ONE

Dear Dr. Hernández-Agüero,

Thank you for submitting your manuscript to PLOS ONE. After careful consideration, we feel that it has merit but does not fully meet PLOS ONE’s publication criteria as it currently stands. Therefore, we invite you to submit a revised version of the manuscript that addresses the points raised during the review process.

As you address the reviewers' comments, we would like for you to attend in particular to:

(1) Reviewer #1's request that you discuss differences in data quality among oases in your analyses and their potential effects on the results; (2) Reviewer #1's request that you amplify the discussion of how temporal scale influences measures of biocultural diversity; and (3) both reviewers’ concern that the cited literature is not the most current / relevant in all cases. 

We look forward to receiving your revised manuscript.

Kind regards,

Raven Garvey, Ph.D.

Academic Editor

PLOS ONE

Journal Requirements:

"Deutsche Forschungsgemeinschaft (DFG; JE 288/9-1, 9-2)"

"We thank Joanna Mitchell and Merve Ören for collecting data on oasis settlements in the Sahara Desert. JMJ acknowledges financial support from the Deutsche Forschungsgemeinschaft (DFG; JE 288/9-1, 9-2)."

"Deutsche Forschungsgemeinschaft (DFG; JE 288/9-1, 9-2)"

6. Please remove your figures from within your manuscript file, leaving only the individual TIFF/EPS image files, uploaded separately. These will be automatically included in the reviewers’ PDF.

Reviewers' comments:

Reviewer's Responses to Questions

**Comments to the Author**

1. Is the manuscript technically sound, and do the data support the conclusions?

Reviewer #1: Yes

Reviewer #2: Yes

2. Has the statistical analysis been performed appropriately and rigorously? 

Reviewer #1: I Don't Know

Reviewer #2: N/A

3. Have the authors made all data underlying the findings in their manuscript fully available?

Reviewer #1: Yes

Reviewer #2: Yes

4. Is the manuscript presented in an intelligible fashion and written in standard English?

Reviewer #1: Yes

Reviewer #2: Yes

5. Review Comments to the Author

Reviewer #1: This is an interesting, well-written and well-structured paper dealing with the biocultural diversity approach which focuses on the co-evolution of biological and cultural systems, using oases in the Sahara Desert as model systems. This work discusses potential drivers and proxies of changing biological and cultural diversity starting from a well-reasoned conceptual framework, and then investigates the biocultural diversity of Algerian oases by applying two complementary methods testing the influence of area and human population.

Linking biological and cultural diversity is a challenging topic especially because of the difficulty of exploring patterns and dynamics of cultural diversity. Nevertheless, this kind of approach is of paramount importance and deserves great attention due to the ongoing globalization processes and environmental degradation that are threatening the biocultural diversity across the world.

The Authors addressed the issue of quantifying biocultural diversity by testing existing (and adapted) indices on data of Sahara oases gathered from an extensive literature review. Overall the results are compelling and indicate that the method is suitable for assessing the biocultural diversity in the Algerian oases. My main concern is that the literature data are not homogeneous for each oasis and therefore some proxy is not properly ‘weighted’ (e.g., in which way the results are influenced by poor data as in the case of oasis Abelassa in the Hoggar group?). I suggest adding a couple of sentences explaining in which way the proposed method overcomes such flaws.

As regards to the conceptual framework, the Authors have the merit of exposing in a clear and scientific way the potential drivers of biological and cultural diversity. Nevertheless, in my opinion the temporal scale – which has an essential role in patterns and trends of biocultural diversity – is not properly introduced. I recommend better explaining why such a ‘deletion’ of a diachronic perspective might matter.

Also, the cited references in the section 3 ‘Linking biological and cultural diversity in oases’ should be improved (e.g., in the par. 3.1.4 the Authors cannot cite only Kuper & Kröpelin 2006 as reference for the settlement history of the Saharan area!). I recommend widen the reference literature of the conceptual framework, so that the well-written text may be improved with relevant and updated reference literature.

The drawing of the follow-up research questions is one of the strong points of the manuscript.

Overall, the topic is in line with the journal and my advice is that the paper needs minor revisions before publication.

Reviewer #2: Dear authors,

your paper is interesting and well structured. THe addressed topic is particulalry relevant for the conservation of oases-related cultural biodiversity.

I only suggest some minor revisions.

The introduction provide a good framework about the issue addressed in the paper, but it would be also interesting to highlight the difference between traditional and modern oases, to clarify the focus of your research.

Lines 76-85. I suggest authors to expand this paragraph with more recent references (the more recent is of 2012) as there are interesting updated studies about oases and related ecosystem services and agro-biodiversity. I.e. studies highlighting that oases far from market centers (as Lybian ones) preserve a higher level of agrobiodiversity.

Lines 265-266. I suggest to include the FAO definition of agrobiodiversity, or at least its citation.

Line 281, not only species, but also varieties, especially if you refer to date palm (1 species, but a lot of varieties!)

Line 530-531. This can be true, but if you state "at present" you should add a "present" reference. Since 2018 (the most updated reference) socio-economic situation could be really different.

6. PLOS authors have the option to publish the peer review history of their article (what does this mean?). If published, this will include your full peer review and any attached files.

Reviewer #1: No

Reviewer #2: No

---

## [Author Response · Author response to Decision Letter 0]

21 Jul 2023

Response to Academic Editor’s comments

R: The manuscript has been revised and format requirements are met. 

"Deutsche Forschungsgemeinschaft (DFG; JE 288/9-1, 9-2)"

R: A Financial Disclosure Statement has been included with information about the role of funders in the study.

"We thank Joanna Mitchell and Merve Ören for collecting data on oasis settlements in the Sahara Desert. JMJ acknowledges financial support from the Deutsche Forschungsgemeinschaft (DFG; JE 288/9-1, 9-2)."

"Deutsche Forschungsgemeinschaft (DFG; JE 288/9-1, 9-2)"

R: A Financial Disclosure Statement has been included with information about the funding including the specific grant number, initials of the author who received the award and the URL of the sponsor’s website. In the acknowledgement statement we added the following sentence: 

L572-573: “We are very grateful to two reviewers and the Editor for providing very helpful comments that considerably improved the text.”

R: Please use Senckenberg as first affiliation and add the billing address: Senckenberganlage 25, 60325 Frankfurt, Germany; Email: generaldirektion@senckenberg.de

R: The map of Figure 1 does not contain images with copyright. We created the map ourselves based on our results. The land outline is taken from Natural Earth (public domain). We included information on trade route nodes from the Old World Trade Routes Project. While all materials on trade routes on the website are published under a Creative Commons BY NC 2.5 license (http://creativecommons.org/licenses/by-nc/2.5/), this license does not apply to the raw data published in the Old World Trade Routes (OWTRAD) Gazetteer (http://www.ciolek.com/OWTRAD/gazetteer-01.html), which is not copyrighted. We have included the trade node points based on the raw coordinates from the Gazetteer, therefore this image can be published under CC BY 4.0. To ensure we cite all the information provided, we added a citation for this information:

L201-202: “based on Old World Trade Routes Project: http://www.ciolek.com/owtrad.html [60]”

And 

L734-735: “Ciolek TM. Old World Trade Routes (OWTRAD) Project. 1999 http://www.ciolek.com/owtrad.html”

6. Please remove your figures from within your manuscript file, leaving only the individual TIFF/EPS image files, uploaded separately. These will be automatically included in the reviewers’ PDF.

R: We have removed the figures from the manuscript.

R: We have included captions for the Figure S1 at the end of the document: 

L841-842 Figure S1: “Relationship between biological and cultural diversity and geographic distances in oases groups in Algeria. Pearson coefficients (r) and significance levels (p) are shown”.

R: All references have been reformatted and double-checked.

Response to reviewers’ comments

Reviewer 1’s comments

Reviewer #1: This is an interesting, well-written and well-structured paper dealing with the biocultural diversity approach which focuses on the co-evolution of biological and cultural systems, using oases in the Sahara Desert as model systems. This work discusses potential drivers and proxies of changing biological and cultural diversity starting from a well-reasoned conceptual framework, and then investigates the biocultural diversity of Algerian oases by applying two complementary methods testing the influence of area and human population.

Linking biological and cultural diversity is a challenging topic especially because of the difficulty of exploring patterns and dynamics of cultural diversity. Nevertheless, this kind of approach is of paramount importance and deserves great attention due to the ongoing globalization processes and environmental degradation that are threatening the biocultural diversity across the world.

The Authors addressed the issue of quantifying biocultural diversity by testing existing (and adapted) indices on data of Sahara oases gathered from an extensive literature review. Overall the results are compelling and indicate that the method is suitable for assessing the biocultural diversity in the Algerian oases. My main concern is that the literature data are not homogeneous for each oasis and therefore some proxy is not properly ‘weighted’ (e.g., in which way the results are influenced by poor data as in the case of oasis Abelassa in the Hoggar group?). I suggest adding a couple of sentences explaining in which way the proposed method overcomes such flaws.

R: We are grateful to the reviewer for the positive and encouraging assessment of the manuscript. However, the review identifies one limitation of the study, and we addressed this by including the following statement: 

L482-491: “A key challenge of the present study was the lack of qualified data and information for the oases in the Sahara Desert. Consequently, we could only extract parts of our ‘proxy wish list’ presented above. In addition, JI considers only pairwise beta diversity and does not provide information on the underlying phenomena such as nestedness (i.e., oases with smaller numbers of species are subsets of oases with higher numbers of species) and turnover (replacement of species by other species). Hence, multiple-site comparisons, including nestedness and turnover, are considered important for future studies [84,92]. In addition, JI is not adjusted for area, which might cause biased results [92]. Lastly, we lack comprehensive data and information to understand and quantify the temporal dynamics of biocultural diversity in the Sahara desert.” 

As regards to the conceptual framework, the Authors have the merit of exposing in a clear and scientific way the potential drivers of biological and cultural diversity. Nevertheless, in my opinion the temporal scale – which has an essential role in patterns and trends of biocultural diversity – is not properly introduced. I recommend better explaining why such a ‘deletion’ of a diachronic perspective might matter.

R: As mentioned above, we could not address the temporal dynamics in our analyses (L326-327 “Here, we provide an analysis of contemporary patterns of Algerian oases, ignoring temporal trends”) due to a lack of comprehensive data and information of those understudied ecosystems. We have added an explanation for this: 

L489-491: “Lastly, we lack comprehensive data and information to understand and quantify the temporal dynamics of biocultural diversity in the Sahara desert.”

Also, the cited references in the section 3 ‘Linking biological and cultural diversity in oases’ should be improved (e.g., in the par. 3.1.4 the Authors cannot cite only Kuper & Kröpelin 2006 as reference for the settlement history of the Saharan area!). I recommend widen the reference literature of the conceptual framework, so that the well-written text may be improved with relevant and updated reference literature.

R: Additional relevant references such as Manning et al. (2013), Marshall & Weissbord (2011) or Manning & Timpson (2014) have been included in the text.

The drawing of the follow-up research questions is one of the strong points of the manuscript.

Overall, the topic is in line with the journal and my advice is that the paper needs minor revisions before publication.

Reviewer 2’s comments

Reviewer #2: Dear authors,

your paper is interesting and well structured. THe addressed topic is particulalry relevant for the conservation of oases-related cultural biodiversity.

I only suggest some minor revisions.

The introduction provide a good framework about the issue addressed in the paper, but it would be also interesting to highlight the difference between traditional and modern oases, to clarify the focus of your research.

R: We are very grateful for the positive assessment of our manuscript and the helpful comments. We have added a sentence mentioning traditional oases at the end of the introduction: 

L74-76: “forming pivotal stepping stones along trade routes and supporting social and economic innovations [27], specially, especially traditional oases [31]”

And in objectives: 

L87-88: “…using traditional oases (sensu [31]) in the Sahara Desert as model systems”

Lines 76-85. I suggest authors to expand this paragraph with more recent references (the more recent is of 2012) as there are interesting updated studies about oases and related ecosystem services and agro-biodiversity. I.e. studies highlighting that oases far from market centers (as Lybian ones) preserve a higher level of agrobiodiversity.

R: We have included more recent references to the paragraph such as Berger et al., 2021 Santoro, 2023 and Santoro et al., 2020. 

Lines 265-266. I suggest to include the FAO definition of agrobiodiversity, or at least its citation.

R: We have added the reference FAO, 1999 to the text:

L262: “…particularly for agrobiodiversity [69–71]”.

Line 281, not only species, but also varieties, especially if you refer to date palm (1 species, but a lot of varieties!)

R: We have included varieties in the text:

 L277-278 “Hence, the number of different cattle and crop species and varieties should be identified for oases”

Line 530-531. This can be true, but if you state "at present" you should add a "present" reference. Since 2018 (the most updated reference) socio-economic situation could be really different.

R: We rephrased this as: 

L530-531 : “Unstable political conditions (e.g., rebel activities) constrain research in many oases of the Sahara Desert [89,101].

---

## [Editor Report · Decision Letter 1]

7 Aug 2023

Oases in the Sahara Desert – Linking Biological and Cultural Diversity

PONE-D-23-03425R1

Dear Dr. Hernández-Agüero,

We’re pleased to inform you that your manuscript has been judged scientifically suitable for publication and will be formally accepted for publication once it meets all outstanding technical requirements.

Kind regards,

Raven Garvey, Ph.D.

Academic Editor

PLOS ONE
---

## [Editor Report · Acceptance letter]

9 Aug 2023

PONE-D-23-03425R1 

­Oases in the Sahara Desert – Linking biological and cultural diversity 

Dear Dr. Hernández-Agüero:

I'm pleased to inform you that your manuscript has been deemed suitable for publication in PLOS ONE. Congratulations! Your manuscript is now with our production department. 

Kind regards, 

on behalf of

Dr Raven Garvey 

Academic Editor

PLOS ONE